# Corneal Transplant Rejections in Patients Receiving Immune Checkpoint Inhibitors

**DOI:** 10.3390/jcm11195647

**Published:** 2022-09-25

**Authors:** Majid Moshirfar, Noor F. Basharat, Tanner S. Seitz, Briana K. Ply, Yasmyne C. Ronquillo, Phillip C. Hoopes

**Affiliations:** 1Hoopes Vision Research Center, Hoopes Vision, 11820 S. State St. #200, Draper, UT 84020, USA; 2John A. Moran Eye Center, University of Utah School of Medicine, Salt Lake City, UT 84132, USA; 3Utah Lions Eye Bank, Murray, UT 84107, USA; 4University of Arizona College of Medicine—Phoenix, Phoenix, AZ 85004, USA; 5Midwestern University Arizona College of Osteopathic Medicine, Glendale, AZ 85308, USA

**Keywords:** cornea, ICI, immune checkpoint inhibitor, monoclonal antibody, rejection, transplant

## Abstract

Immune checkpoint inhibitors (ICIs) are antibodies that target and block immune checkpoints. These biologics were initially approved by the United States Food and Drug Administration (US FDA) in 2011 for the management of melanoma. Since then, the use of ICI therapy has increased, with many new medications on the market that treat approximately 50 types of cancers. Patients receiving this therapy are at an increased risk for transplant rejection, including corneal rejection. Ophthalmologists must be aware of individuals receiving ICI therapy as it may be a relative contraindication for patients with a history of corneal transplantation. Patients on ICIs may also experience ocular side effects, including uveitis, dry eye, and inflammation, while on checkpoint inhibitor therapy. This commentary discusses the current understanding of immune checkpoint inhibitors, their mechanism of action, their ocular side effects, and their role in corneal transplant rejection.

Immune checkpoint inhibitors (ICIs) are monoclonal antibodies commonly used in the treatment of approximately 50 cancer types including melanoma, bladder, kidney, lung, liver, Merkel cell, and other skin cancers [1]. ICIs revolutionized the fields of chemotherapy and immunotherapy [1] as they introduced a novel mechanism to treat cancer with a less toxic adverse effect profile [2]. The first immune checkpoint inhibitor, ipilimumab (Yervoy), was U.S. FDA approved in 2011 for the treatment of advanced melanoma [3]. Since ipilimumab entered the market, eight more checkpoint inhibitors have become available [4,5].

Immune checkpoints involve stimulatory and inhibitory pathways that affect immune cells and maintain a balance between pro- and anti-inflammatory signaling [6]. Cancer cells stimulate inhibitory immune checkpoints leading to a reduction in the normal immune response; thus, the mechanism to suppress cancerous cells is no longer functional [7]. Because the cancer cells have escaped the immune response, they can continue to grow. ICIs function by blocking the effects of specific inhibitory pathways [4]. Because of this overactive immune response, the immune system can target antigenic cells, including cancerous cells and transplanted donor tissue. 

Inhibition of the T cell response, which is a target of CTLA-4 and PD-1 inhibitors, is thought to be responsible for the donor transplant rejections, including corneal tissue. CTLA-4 is upregulated on active T cells and competes with the CD28 costimulatory molecule on T cells to bind to the B7-1 and B7-2 ligands on the antigen-presenting cell (APC) in order to inhibit T cell receptor (TCR) signaling. This ultimately suppresses CD-28 mediated T cell activation (Figure 1) [8,9]. In addition to CTLA-4, PD-1 is upregulated on active T cells. PD-1 binds to its ligand, PD-L1, on the APC and inhibits T cell activation (Figure 2) [9]. Inhibition of these molecules and other immune checkpoints, including but not limited to Lag-3, Tim-3, Tigit, and Vista [9], results in an immune-mediated response involving T cell overactivity [2]. These T cells undergo clonal proliferation and form CD4^+^ T helper cells [10]. Through various mechanisms, CD4^+^ cells cause a delayed-type hypersensitivity immune response directed at the alloantigens on the graft [10]. As the name suggests, these alloantigens are antigenically different, like cancer cells, so the recipient perceives them as foreign [11,12]. This immune response destroys the allogeneic/donor transplant tissue, including corneal tissue [10].

Eye care professionals may encounter patients who experience ocular changes while on ICI therapy. The spectrum of toxicity of the ICIs is referred to as immune-related adverse events (irAEs) [13]. Ocular side effects or irAEs, including conjunctivitis, uveitis, dry eye, ocular myasthenia gravis, retinal detachment, uveal effusion, and inflammation, have been reported in patients on atezolizumab, avelumab, cemiplimab, durvalumab, ipilimumab, nivolumab, and pembrolizumab [13,14]. A unique complication, Vogt–Koyanagi–Harada disease-like pan-uveitis, has also been seen in a patient on nivolumab therapy [15]. Resolution of symptoms often occurs after administration of corticosteroids [16]. Individuals presenting with these side effects should prompt eye care professionals to inquire about ICI use, especially those with a history of cancer. 

ICIs have been implicated in solid organ transplant rejection, including the heart, liver, and kidneys [17,18,19,20]. While studies are lacking regarding all ICIs and transplant surgery, CTLA-4 medications have shown lower rates of rejection than the PD-1/PD-L1 inhibitors [21]. The use of PD-1/PD-L1 agents, especially nivolumab and pembrolizumab, is contraindicated in solid organ transplants owing to high rejection rates [22]. A hypothesized cause of elevated transplant rejection rates while on ICI therapy is the inhibition of PD-1/PD-L1 in host T-cells and graft dendritic cells [23]. Thus, ICIs are relatively contraindicated for organ transplant recipients.

PD-L1 is highly expressed within the corneal endothelium, allowing for increased interaction between PD-L1 and ICIs [24]. Both PD-1 and PD-L1 inhibitors have consistently been associated with higher rates of corneal rejection [22]. In mice injected with antibodies against PD-1, the rate of rejection in corneal allografts was 100% [25,26]. The average rate of solid transplant rejection in the United States general population is 10–20%, while patients on ICI therapy approach 41% [27]. In general, the rate of corneal rejection ranges from 2.3% to 68% [28]. Although the incidence of corneal transplant rejection in patients using ICIs is relatively low, ophthalmologists must consider the risk of corneal rejection as increased rates have been reported in other solid organ transplant procedures [29].

Over the years, there has been some correlation between ICIs and corneal rejection. In one case report, an 85-year-old woman with worsening visual acuity experienced corneal rejection after initiating nivolumab therapy one year after her corneal transplant [30]. A second report of corneal rejection occurred in a woman three months after initiating pembrolizumab for urothelial cell carcinoma [31]. Signs of transplant rejection include worsening visual acuity, corneal edema, corneal vascularization, and stromal or subepithelial infiltrates. The evaluation and management of patients on ICI therapy presenting with signs of acute rejection includes a thorough slit lamp examination, corneal thickness measurements, and corticosteroid administration [16,32]. One study suggests that taking the risk with ICIs may be worthwhile if the patient has had a corneal transplant in one eye, but good vision in the other [25]. However, the decision to start a patient on ICIs may require more scrutiny in those with bilateral transplants or monocular patients with good vision only in the eye with the corneal graft as there is a greater risk associated with these conditions. 

After beginning checkpoint inhibitor therapy, a baseline eye examination is recommended in corneal graft patients [31]. Consultation is encouraged between the ophthalmologist and the managing oncologist regarding patient disease and the risks of medication discontinuation. In patients undergoing ICI therapy, corneal transplantation and potential rejection should be weighed carefully, considering the general medical need of the patient versus the risk of visual impairment. 


**Conclusions**


Immune checkpoint inhibitors are monoclonal antibodies indicated for solid tumor treatment. Since 2011, ICIs have become first- or second-line therapeutics for cancer management. Patients receiving this therapy may experience ocular side effects, including uveitis, dry eye, ocular myasthenia gravis, and inflammation. Management of these symptoms typically consists of topical or systemic corticosteroids. A more severe potential complication of these medications is transplant rejection, including corneal rejection. Ocular management of patients with a history of corneal transplants is recommended before and after beginning ICI therapy and should include a comprehensive eye examination, detailed corneal assessment, and collaboration with the patient’s primary care physician and oncologist for prevention and treatment of any adverse effects. 

## Figures and Tables

**Figure 1 jcm-11-05647-f001:**
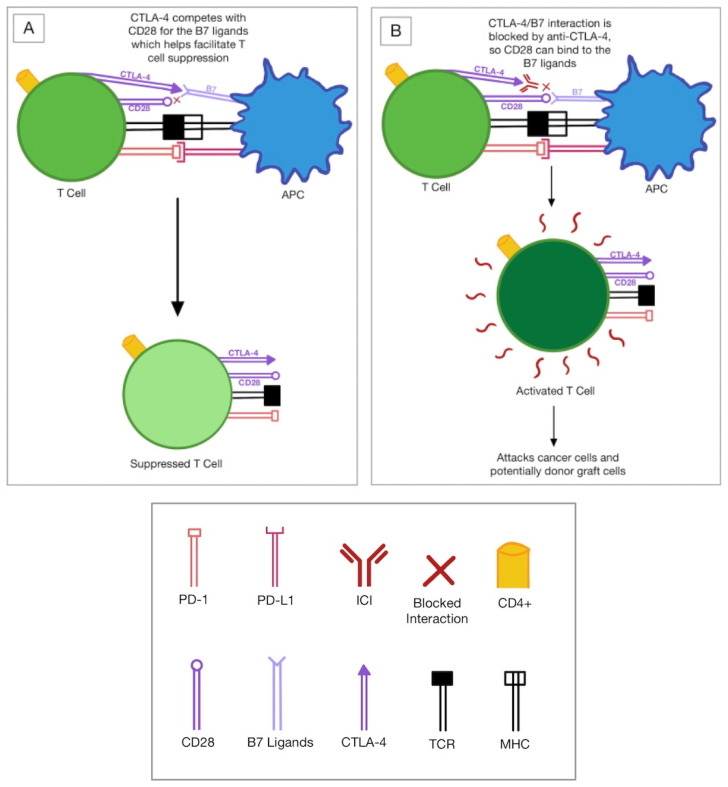
(**A**) No ICI (**B**) CTLA-4 inhibitor mechanism. Abbreviations: APC: antigen-presenting cell; PD-1: programmed death-1; PD-L1: programmed death ligand-1; CTLA-4: cytotoxic T lymphocyte-associated molecule-4; CD28: cluster of differentiation 28; ICI: immune checkpoint inhibitor; MHC: major histocompatibility complex; TCR: T cell receptor.

**Figure 2 jcm-11-05647-f002:**
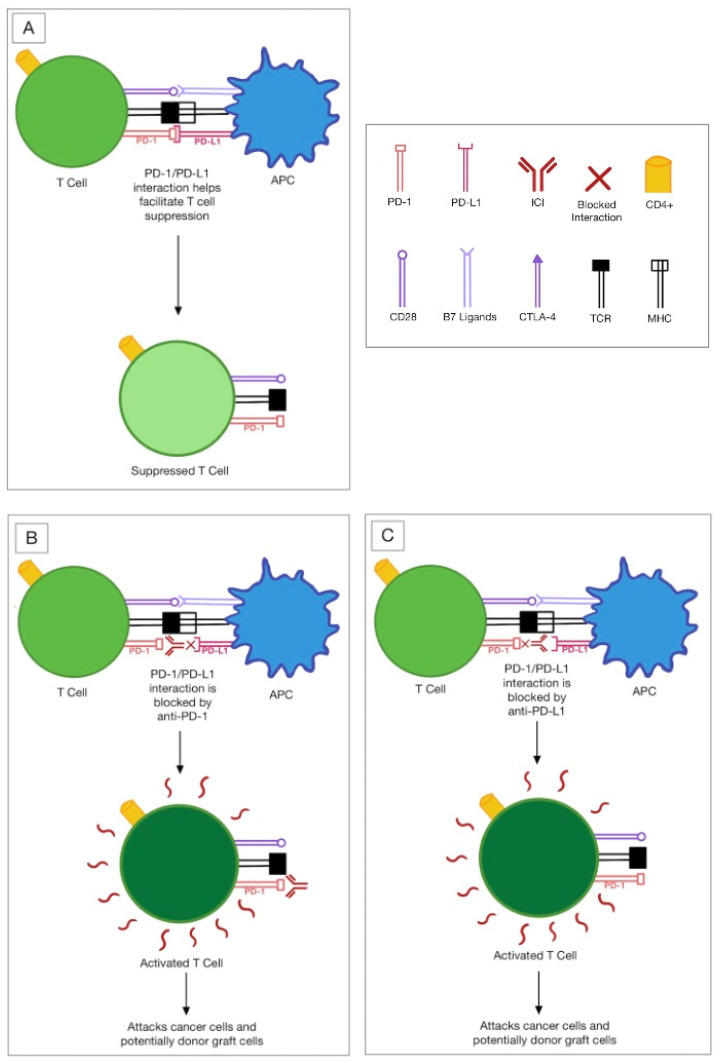
(**A**) No ICI (**B**) PD-1 inhibitor mechanism (**C**) PD-L1 inhibitor mechanism. Abbreviations: APC: antigen-presenting cell; PD-1: programmed death-1; PD-L1: programmed death ligand-1; CTLA-4: cytotoxic T lymphocyte-associated molecule-4; CD28: cluster of differentiation 28; ICI: immune checkpoint inhibitor; MHC: major histocompatibility complex; TCR: T cell receptor.

## Data Availability

Data sharing is not applicable to this article as no datasets were generated or analyzed during the current study.

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
