# Peer review of "Corneal Transplant Rejections in Patients Receiving Immune Checkpoint Inhibitors"

_jcm, 2022, doi:10.3390/jcm11195647_

Round 1

Reviewer 1 Report

The authors present a fairly interesting and relevant for both oncologists  and ophthalmologists mini review on a class of new drugs that are associated with various ocular side effects. 

The paper is a Mini review that provides some basic information about immune checkpoint inhibitors, new drug class used in oncologists patients.

Furthermore, they mention few case reports that suggest an association with the drug class under review.

I did not detect noteworthy strengths or weaknesses that could help the authors improve their paper. 

It is short and informative and due to its topic and fair quality I find it is worth publishing. 

Author Response

Thank you for your comments!

Reviewer 2 Report

ICI review (Morshifar)
  • Is it necessary to include Table 1? Perhaps it can be in a supplement, as it does not add to the substance of the paper
  • Paragraph 3 is confusing and we only find out later in the paragraph why this is relevant to the reader (T cell responses may be responsible for destroying donor transplant tissues). The first line of this paragraph should say exactly that, "Inhibition of the T cell response, which are targets of CTLA-4 and PD-1 inhibitors, is thought to be responsible for the donor transplant rejections, including corneal tissue." I would delete the first sentence as it is now (line 42-43) as it is not adding much, and then can describe in more detail how the drugs work (lines 44-52).
  • Figure 2- bottom images B and C are basically identical. Can you just write, "PD-1/PD-L1 interaction is blocked by anti-PD 1 or anti-PD L1" to reduce redundancy?
  • Paragraph four seems out of place. The rest of the paper focuses on immunology and the mechanism of corneal transplant rejection, whereas this paragraph only talks about other ocular changes. I would just take this paragraph out all together.
  • Switch order of paragraphs 6 and 7.  The flow is such that it makes sense to focus on the immunology, then anatomy, and then clinical vignettes.
  • The sentences, "However, signif-102 icant advances in corneal transplant surgery have been made regarding endothelial trans-103 plantation. Techniques such as Descemet’s stripping endothelial keratoplasty (DSEK), 104 Descemet’s membrane endothelial keratoplasty (DMEK), and deep anterior lamellar ker-105 atoplasty (DALK) involve less tissue, potentially decreasing the risk of rejection. Possible 106 mechanisms for decreased risk include minimized PD-L1 exposure and less immunogen-107 icity introduced from person to person " do not make sense in this paragraph and should be removed. If endothelial rejection is the primary type of rejection, then replacing endothelial tissue is not going to reduce the rate of this type of rejection.
  • Line 109/110: is there a known rejection rate of corneal tissue? If so, please state it- if it is unknown, please state that as well.
  • Last sentence: this is perhaps too strongly worded.  should it be a consideration in patients who are monocular? I would soften the tone a bit 
